# Mucosal Immunity and the Gut-Microbiota-Brain-Axis in Neuroimmune Disease

**DOI:** 10.3390/ijms232113328

**Published:** 2022-11-01

**Authors:** Kathryn G. Sterling, Griffin Kutler Dodd, Shatha Alhamdi, Peter G. Asimenios, Ruben K. Dagda, Kenny L. De Meirleir, Dorothy Hudig, Vincent C. Lombardi

**Affiliations:** 1Department of Biology, University of Nevada, Reno, NV 89557, USA; 2Department of Microbiology and Immunology, University of Nevada, Reno School of Medicine, Reno, NV 89557, USA; 3Clinical Immunology and Allergy Division, Department of Pediatrics, King Abdullah Specialist Children’s Hospital, King Saud bin Abdulaziz University for Health Sciences, Ministry of National Guard Health Affairs, Riyadh 11426, Saudi Arabia; 4Department of Pharmacology, School of Medicine, University of Nevada, Reno, NV 89557, USA; 5Himmunitas Foundation, 2800 Mechelen, Belgium

**Keywords:** sIgA, Alzheimer’s disease, autism, microbiome, mucosal immunity, myalgic encephalomyelitis, Parkinson’s disease, multiple sclerosis, Huntington’s disease

## Abstract

Recent advances in next-generation sequencing (NGS) technologies have opened the door to a wellspring of information regarding the composition of the gut microbiota. Leveraging NGS technology, early metagenomic studies revealed that several diseases, such as Alzheimer’s disease, Parkinson’s disease, autism, and myalgic encephalomyelitis, are characterized by alterations in the diversity of gut-associated microbes. More recently, interest has shifted toward understanding how these microbes impact their host, with a special emphasis on their interactions with the brain. Such interactions typically occur either systemically, through the production of small molecules in the gut that are released into circulation, or through signaling via the vagus nerves which directly connect the enteric nervous system to the central nervous system. Collectively, this system of communication is now commonly referred to as the gut-microbiota-brain axis. While equally important, little attention has focused on the causes of the alterations in the composition of gut microbiota. Although several factors can contribute, mucosal immunity plays a significant role in shaping the microbiota in both healthy individuals and in association with several diseases. The purpose of this review is to provide a brief overview of the components of mucosal immunity that impact the gut microbiota and then discuss how altered immunological conditions may shape the gut microbiota and consequently affect neuroimmune diseases, using a select group of common neuroimmune diseases as examples.

## 1. Introduction

The gastrointestinal (GI) tract is home to trillions of bacteria that represent hundreds, if not thousands of discrete species. For many years, it was generally assumed that these bacteria did not impart any significant health benefit to the host, with the notable exception of competing with pathogenic bacteria for nutrients and colonization sites. This assumption was largely based on observations that individuals who receive long-term antibiotic therapy often develop pathogenic and life-threatening *C. difficile* infections as a result of the loss of commensal bacteria [1]. It is now universally accepted that the gut microbiota contributes to overall health at several levels. For instance, the fermentation of fiber by the microbiota produces short-chain fatty acids (SCFAs), such as acetate, butyrate, and propionate, all of which have several health benefits. Indeed, butyrate is the preferred fuel source for colonic epithelial cells (colonocytes); however, in the absence of dietary fiber, and in turn the absence of butyrate, colonocytes utilize an alternative energy source: the protective high molecular weight glycoproteins found in GI mucus [2]. This consumption of glycoproteins can lead to a compromised epithelial barrier, bacterial translocation across the gut epithelium, and ultimately, systemic inflammation [3,4,5]. Butyrate downregulates nuclear factor-kappa B (NF-kB) [6] and is also a potent histone deacetylase (HDAC) inhibitor [7], which is largely responsible for the anticancer benefits of a high-fiber diet [8,9]. 

In addition to producing molecules that maintain homeostasis, the gut microbiota also synthesizes or promotes the synthesis of metabolites that impact neurological function and plays a critical role in cognition, learning, and memory. For example, 5-hydroxytryptamine (serotonin) is primarily produced by enterochromaffin cells in the gut [10], and its production is heavily influenced by commensal bacteria [11,12]. Conversely, alterations in the gut microbiota (known as dysbiosis) can lead to lipopolysaccharide (LPS), an endotoxin derived from the outer membrane of Gram-negative bacteria, crossing the gut epithelial barrier, and promoting systemic inflammation [3,4,5]. Previous studies conducted in animal models (e.g., the 5XFAD murine model of Alzheimer’s disease) suggest that LPS also increases the permeability of the blood–brain barrier, thus representing a connection between gut dysbiosis and neuroinflammation [13]. The microbial small molecules that enter the blood, including butyrate and LPS, can then cross the blood–brain barrier. Although the molecular crosstalk between the gut and brain is beyond the scope of this review, the reader is directed to the following excellent reviews that address this topic, written by Chen et al. [14], Parker et al. [15] and Gwak et al. [16]. 

In this review, we provide a brief overview of the architecture of the mucosal immune system, followed by a description of its components, and then discuss how the dysregulation of the mucosal immunity affects six major diseases: Alzheimer’s, autism, Parkinson’s, multiple sclerosis, Huntington’s disease, and myalgic encephalitis. 

## 2. Architecture of the Gastrointestinal Immune System

The gut is often referred to as the largest immune organ in the body, and although it is not a discrete immune organ, its significant impact on host immunity can be appreciated when one considers that the small intestine is home to the majority of the body’s T cells and has a surface area of approximately 400 square meters [17]. The GI immune system has a specific architecture, comprised of organized tissue as well as connective (scattered) lymphoid tissue where both innate and adaptive immune cells are found. The organized tissue is where naïve T and B cells interact with antigen-presenting dendritic cells (DCs) and is referred to as gut-associated lymphoid tissue (GALT). This includes the Peyer’s patches (PPs), isolated lymphoid follicles (ILF), and the appendix. Intraepithelial lymphocytes (IELs) are scattered among the intestinal epithelial cells (IECs) that form a physical barrier between the host tissue and commensal bacteria. The IELs are mostly composed of activated T cells. Throughout the connective tissue of the lamina propria (LP), which is just below the gut epithelium, is a mix of innate and activated adaptive immune cells, including T and B cells, plasma cells, eosinophils, macrophages, and mast cells [18,19]. The GALT and scattered lymphoid tissue drain through a system of lymphatics into the mesenteric lymph nodes (MLNs). A summary of the GALT of the small intestine is illustrated in Figure 1. Although the MLNs are not technically considered a part of the GALT, we include MLNs because of their importance in GI mucosal immunity. Furthermore, while the tonsils and adenoids in the throat are frequently considered part of the GALT, for the purposes of this review they are excluded. 

### 2.1. Intestinal Epithelial Cells (IECs)

The IECs are the principal point of contact between the host and the microbes within the intestinal lumen. The IECs of the small intestine are primarily composed of columnar epithelial cells called enterocytes that line the surface of villi, finger-like processes that line the intestine. Interspersed among these enterocytes are the IELs, which are almost exclusively T cells and are mainly cytotoxic T cells (CD8+) or gamma-delta (γδ+) T cells [21,22]. 

### 2.2. Lamina Propria (LP)

The connective tissue directly beneath the IECs is the LP, which is home to innate and adaptive immune cells, including macrophages, mast cells, eosinophils, DCs, T cells, B cells, plasma cells, and innate lymphoid cells (ILCs), all of which are scattered throughout the tissue. Mononuclear cells are the most prevalent immune cell in the LP, with greater than 50% being T and B cells, followed by LP plasma cells that primarily produce dimeric immunoglobin A (IgA) and pentameric IgM [23]. Within the LP are organized structures called ILFs that facilitate the sampling of luminal antigens necessary to elicit adaptive immune responses. These follicles contain primarily CD4+ T cells as well as B cells, which can make all classes of immunoglobulins [24].

### 2.3. Peyer’s Patches (PPs)

PPs are located primarily in the duodenum and ileum but are found sporadically in the jejunum and are absent from the colon [25]. PPs contain distinctive areas; the outermost is a thin single-cell layer of epithelium known as the follicle-associated epithelium (FAE). Directly below the FAE is the subepithelial dome, which is rich in T cells, B cells, macrophages, and DCs. Below the subepithelial dome are the follicular and the interfollicular areas, which contain B-cell follicles, each surrounded by a T-cell zone [26,27]. Analogous to peripheral lymph nodes, PPs also contain germinal centers, which are composed of proliferating B cells that ultimately differentiate into plasma cells that primarily produce IgA and to a lesser but significant extent, IgM [28]. 

As in peripheral lymph nodes, T and B cells enter PPs by crossing the walls of high endothelial venules (HEVs) through the engagement of the chemokine receptor CCR7 [29]. In contrast to peripheral lymph nodes, PPs do not have afferent lymphatics, so antigen enters directly from the gut across specialized epithelium made up of unique cells known as microfold cells (also known as M cells) [30]. Antigens are transported from the gut lumen by M cells, where the antigens access antigen-presenting cells (APCs), primarily conventional DCs (cDCs) and B cells, and to a lesser extent, plasmacytoid DCs (pDCs) and macrophages. An important feature of M cells is the selective access they create for antigens to cross from the intestinal epithelium into the PP. Several intestinal M-cell receptors participate in the antigen-uptake process; for example, the glycosylphosphatidylinositol (GPI)-anchored protein GP2 binds to type I pili-expressing bacteria (such as *E. coli* and *S. typhimurium*) and actively transports the bacterial antigens to the basolateral side of the epithelium via transcytosis [31]. The GPI-anchored protein uromodulin facilitates the uptake of *L. acidophilus* through its interaction with bacterial surface layer protein A [32]. M cell receptors also allow pathogenic bacterial products to penetrate the protective epithelium; for example, GP2 also binds hemagglutinin A1 of botulinum neurotoxin, allowing botulism toxin to enter the body [33]. Pathogens including poliovirus, *Salmonella typhimurium*, *Salmonella typhi, Yersinia enterocolitica*, and *Vibrio cholera* all adhere selectively to M cells to gain access across the epithelium [34,35,36,37,38]. The activity of PPs profoundly affects the immune control of microbes and limits systemic access to microbial products. The number of PPs increases until 15–20 years of age and then decreases by 50% as people age [26,39]. 

### 2.4. Mesenteric Lymph Nodes and Lymphatics

The MLNs are located at the mesenteric root and throughout the mesentery. They are the draining lymph nodes of the gut, with the IECs, LPs, and PPs all connected by efferent lymphatic vessels that drain to MLNs. Lymphocytes that encounter antigens in the PPs proliferate and commence differentiation into mature antigen-specific effector cells and subsequently migrate to the MLNs, where they undergo their final maturation [40]. Lymphocytes that were primed in the gut can enter the systemic circulation and use specific combinations of integrins and chemokine receptors to home back to the gut and their specific target tissues (see Section 4.1). 

The MLNs serve a unique function in the induction of small intestinal tolerance to food proteins (oral tolerance). Oral tolerance prevents potentially antigenic substances from eliciting a cellular or humoral immune response [41]. Although still not well understood, current data suggest that the induction of oral tolerance relies on several mechanisms, including T-cell anergy and/or deletion and the activation of antigen-specific regulatory T cells (Tregs) that secrete interleukin (IL)-10 [42,43]. DCs also participate in oral tolerance. The DCs that migrate to MLNs express the homing receptor CCR7 as well as the DC markers CD11b and CX3CR1 or express CD11c, CD103, and CX3CR1 [44,45]. In contrast, tolerogenic DCs resident in the LP express CD11b and CX3CR1 but do not express CCR7. Moreover, these DCs also express the immunosuppressive cytokine IL-10 [46]. Whereas tolerance to consumed antigens is primarily restricted to the small intestine, tolerance to microbial antigens is mostly restricted to the terminal ileum or colon [47,48]. 

### 2.5. Appendix

The appendix is a finger-shaped projection that sits at the junction of the small intestine and large intestine and is posterior-medially connected to the cecum. Though once thought to be vestigial, recent clinical and phylogenetic studies suggest that the appendix is a part of the mucosal immune system [49,50]. Bollinger et al. proposed that the appendix functions as a reservoir for beneficial gut bacteria for times when GI illness flushes the intestine of commensals [49]. In 2011, in a retrospective, multivariate study, Im et al. reported that the presence of an appendix was inversely associated with the recurrence of *C. difficile* infections (*p* < 0.0001; adjusted relative risk, 0.398) [51]. However, this hypothesis was challenged by studies where a positive correlation between the presence of an appendix and *C. difficile* infection was observed [52]. Whether or not an appendectomy is a predisposing factor to *C. difficile* infection, the appendix has the histological structure of other GALT, including mucosa, submucosa, and muscularis externa and serosa, and supports its own commensal microbial population [53]. As a result, a localized microbial population would have the potential to generate its products.

## 3. Innate Immunity at the GI Mucosal Surface

### 3.1. Mucus Production by the Intestinal Epithelium

A remarkable trait of the intestinal epithelium is its prodigious regenerative capacity. Stem cells residing within pockets of the villi called crypts, continuously replenish the epithelium, allowing it to regenerate the entire luminal lining every five to seven days [54,55]. At the base of these crypts are Paneth cells that specialize in the production of antimicrobial peptides and enzymes, including defensins, lysozyme, and secretory phospholipase A_2_, all of which impact the composition of the microbiota [56].

The intestinal epithelium fulfills its protective function largely through the production and secretion of mucus. Attached to the outer surface of IECs is a filamentous coat of weakly acidic sulfated mucopolysaccharides called the glycocalyx [57]. Interspersed among the columnar enterocytes are goblet cells, which secrete thick viscous mucus, forming a protective barrier that lines the top of the glycocalyx [58]. The mucus serves two important functions; first, it provides lubrication for digesting materials moving through the gut. Second, it creates a protective barrier, decreasing bacterial interaction with the IECs [59,60]. The mucus layer also provides a scaffold for the retention of antimicrobial peptides, as well as secretory IgA (sIgA), which binds to mucus through its interaction with carbohydrate elements of the secretory component, a protein that is added to IgA when secretory sIgA is formed [61]. In turn, sIgA binds to commensal bacteria and prevents their translocation across the epithelium; this process is known as immune exclusion (see Section 4.2) [62]. As most of the host-bacteria communication occurs at the mucus interface, abnormal IgA or mucus production or composition can facilitate disease by allowing the invasion of the intestinal epithelium or by disrupting the local microbiota. These abnormalities have been demonstrated in various neurological diseases, including Alzheimer’s disease (AD), Parkinson’s disease (PD), autism, and HIV-associated neurological diseases [63,64]. 

Tight junctions of the epithelium limit immune challenges. The junctions form a barrier between the plasma membranes of adjacent cells, inhibiting the translocation of pathogens as well as regulating intestinal permeability by ensuring that materials that cross epithelium do so only by passing through epithelial cells in a controlled manner [65]. Disruptions and alterations to the integrity of tight junctions of the intestinal epithelium have been reported in those with PD [66] and children with autism [67]. Moreover, alterations in the intestinal epithelium and tight junction integrity have been associated with a less diverse microbiota, and with an altered relative abundance in several taxa [68]. 

### 3.2. Pattern Recognition Receptors and Innate Lymphoid GI Cells

IECs express a variety of pattern recognition receptors (PRRs) that bind microbial molecular motifs associated with danger to the host (Figure 2); such motifs are commonly referred to as pathogen-associated molecular patterns (PAMPs). Examples of common transmembrane PRRs include the Toll-like receptors (TLRs) 1–10 (in humans) [69], and the C-type lectin receptors (dectin-1, dectin-2, dectin-3, DC-SIGN, mannose-binding lectin, and mincle) [70]. TLRs largely respond to microbial-derived nucleic acids and bacterial-derived molecules, such as lipopeptides, LPS, and bacterial flagellins. Conversely, C-type lectin receptors respond to pathogen-derived carbohydrates, such as β-glucan, α-mannan, and fructose [71]. The cytosolic PRRs include but are not limited to the nucleotide-binding oligomerization domain (NOD) proteins, the nod-like receptors proteins (NLRPs), RIG-I, MDA-5, AIM2, and LGP2 [72]. 

Activation of PRRs initiates innate immune responses through the production of cytokines and chemokines as well as providing co-stimulation for antigen-presenting cells (APCs). In the gut, PRRs play a critical role in regulating the composition of the resident microbiota. PRRs initiate inflammatory conditions in response to pathogens with a concurrent reduction in the inflammatory responses to benign commensal bacteria. PRRs also increase the amount of sIgA produced by triggering immunoglobulin isotype switching in B cells and promoting cell proliferation in response to mucosal damage [73]. In addition to being expressed by IECs, PRRs are also expressed by most APCs, including cDCs, pDCs, macrophages, and B cells [72,74,75] throughout the body, which suggests that when microbial products are widespread, PRRs can occur in many neuronal sites [76,77,78,79]. 

Inflammasomes, which are activated in response to PRR engagement, are multiprotein complexes found in the cytosol. Their assembly begins through the activation of the intracellular NOD-like receptors NLRP1, NLRP3, NLRC4, or AIM2 [80,81] which all respond to microbial products that are within the cytoplasm. In the gut, dysfunctional inflammasome formation is linked to immunopathology in the gut–brain axis, chronic gut inflammation, and impaired tumor immunosurveillance [82].

**Figure 2 ijms-23-13328-f002:**
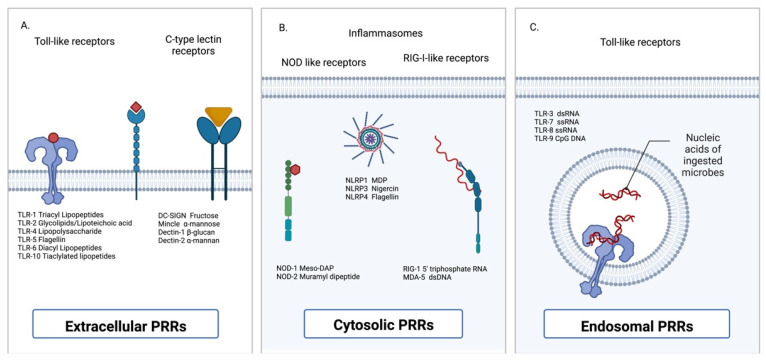
Cellular localization of pattern recognition receptors. (**A**) Extracellular pattern-recognition receptors (PRRs) include but are not limited to the Toll-like receptors (TLRs) and the C-type lectin receptors. (**B**) Cytosolic PRRs include the nucleotide-binding oligomerization domain (NOD)-like receptors and the retinoic acid-inducible gene 1 (RIG-I)-like receptors. (**C**) Endosomally-located PRRs are restricted to the TLRs [83].

Although the activation of PPRs in non-GI tissue typically results in an inflammatory response, this is not always the case in the gut, nor is inflammation by the inflammasome-products IL-1β and IL-18, guaranteed. Indeed, activation of PPRs in the gut usually produces a tolerogenic response. Previous studies have shown that NLRP3 promotes intestinal barrier integrity via the expression of IL-1β and IL-18, which inhibits the infiltration of granulocytes [84]. The importance of NOD-like receptors in maintaining gut homeostasis is underscored by the number of NLP-associated proteins that are genetically associated with susceptibility to inflammatory bowel disease. In fact, a *NOD2* polymorphism was identified in 2001 as the first susceptibility locus for Crohn’s disease and subsequently, at least 60 additional variants in this gene have been associated with this disorder [85,86,87,88]. 

### 3.3. Antimicrobial Peptides of the GI Tract

Antimicrobial peptides (AMP) are amphipathic proteins secreted onto mucosal surfaces and function to kill pathogenic bacteria and fungi by forming pores in cellular membranes [89]. AMPs also play an important role in regulating the interactions between commensal microbes and host tissues [90]. Three common families include defensins, cathelicidins, and histatins. In response to cytokine signaling, Paneth cells, which produce the α-defensins known as cryptdins and RegIIIγ, are the principal producers of antimicrobial peptides in the gut [91]. Defensins act on a broad range of microbes, and function in immunoregulation by promoting or inhibiting inflammatory response [92,93]. Cathelicidins, which are produced in an inactive form and stored in cytoplasmic granules of neutrophils and macrophages, are activated and released into phagosomes in response to infection [94]. Lastly, histatins are constitutively produced in the salivary glands and inhibit the growth of fungi [95]. A 2021 publication raises the issue of antimicrobial peptides also having amyloidogenic activity, which would be an unanticipated means by which infections could affect AD and PD [96]. 

### 3.4. Populations of Innate Lymphoid Cells in the GI Tract

Innate lymphoid cells (ILCs) are tissue-resident immune cells (reviewed, in [97]) that develop from common lymphocyte progenitors that express the PLZF transcription factor, which commits the cells into ILC progenitors [98]. The primary function of ILCs is to potentiate innate immune responses. ILCs are divided into groups based on the cytokines they produce (Figure 3), and the transcription factors involved in their development. Group 1 ILCs express interferon γ in response to IL-12 and IL-18, thus promoting an antiviral state [99]. Group 2 ILCs secrete IL-4, IL-5, IL-9, and IL-13 to improve mucosal and antiparasitic immunity in response to thymic stromal lymphopoietin, IL-25, and IL-33 [100,101]. Group 3 ILCs are mostly concentrated in the gut and produce IL-22 and IL-26 in response to IL-23 [100,101]. IL-22 induces the expression of antimicrobial peptides and its signaling is also critical for the fucosylation of glycoproteins and glycolipids in the gut, which provides an energy source for the resident microbiota [102,103]. A subsection of group 3 ILCs are lymphoid tissue inducer cells (LTi), which contain the nuclear receptor RORγ(t) [104]. These cells initiate the organogenesis of lymph nodes and PPs in the gut [104]. Group 4 is often assigned to natural killer (NK) cells, which are discussed later in the review. The four ILC groups have parallel functions with effector T cells based on cytokine production, and each ILC has a T cell counterpart that produces similar cytokines: Th1/ILC1, Th2/ILC2, TH17/ILC3, and cytotoxic NK cells/CD8 T cells. Based on these similarities, ILC pathogen-associated responses can be accepted to orchestrate immune responses and affect the balance of microbes in the GI. 

ILCs in the gut indirectly interact with the resident microbiota through cytokine orchestration of immunity; likewise, the microbiota influences the differentiation of intestinal ILCs. Indeed, ILCs can have a protective or harmful effect on the microbiota depending on the circumstances [106]. ILCs can also be activated through neurotrophic factor signals from glial cells of the enteric nervous system, to induce ILC3 release of cytokines and increase inflammation [106]. Both immunity and inflammation work to maintain gut homeostasis by regulating the resident microbiota and raising or lowering the inflammatory damage [106]. 

## 4. GI Adaptive Immunity

### 4.1. T Cell Subsets

The GALT contains the largest population of T cells in the body. T cells differentiate in the thymus and are identified by their expression of the surface protein CD3 and subcategorized by the expression of their T cell receptors (TCRs), co-receptors, transcription factors, and secreted cytokines. Like peripheral T cells, gut-associated CD4+ T helper cells recognize antigens presented by major histocompatibility protein (MHC) class II molecules, whereas cytotoxic CD8+ T cells recognize antigens presented on MHC class I. Two major populations of T cells, distinguished by the absence or presence of coreceptor CD8α/α, are found in the gut. Conventional intestinal T cells express the TCR α/β along with either co-receptors CD4 or CD8α/β. T-helper cells, stimulate B cells to secrete antibody, and stimulates macrophages to kill ingested microbes, but also help activate cytotoxic T cells to kill infected target cells. The other population, which is commonly referred to as nonconventional intestinal T cells, expresses either TCRα/β or TCRγ/δ (commonly referred to as γ/δ T cells) and usually the coreceptor CD8α/α. CD8α/α T cells are derived from thymic IEL precursors [107,108], and studies conducted in animal models suggest that CD8α/α engagement negatively regulates TCR activation [109]. Most γ/δ T cells in the gut reside in the intraepithelial layer where they regulate immunosuppressive functions of other IELs as well as promote tolerance (e.g., of food antigens and commensal bacteria) and healing of epithelial tissue [110]. In humans, as much as 40% of the resident intestinal T cells in a noninflamed large intestine are γ/δ T cells [111]. Although the functions of these γ/δ T cells are still to be fully resolved, it is believed they have both innate and adaptive characteristics. Their TCRs display PRR characteristics and thus can respond quickly to infection. 

In addition to the intraepithelial layer, the LP is replete with T cells. Most T cells found in the LP, as well as in the epithelium, migrated to the gut upon T cell expression of gut homing receptors [112,113,114]. Naïve T cells initially leave the thymus and migrate to the GALT, where they are primed by APCs. Once primed in the gut, they will upregulate α4β7 integrin as well as the adhesion molecule LFA-1, which allows them to later home back to the gut after they re-enter the peripheral circulation [115]. Furthermore, T cells that are initially primed in the small intestine upregulate CCR9, which binds the chemokine CCL25. This chemokine is constitutively produced by the small intestine, thus allowing the cells to follow the chemokine gradient back to the sites where they were activated. Similarly, T cells initially primed in the large intestine upregulate the receptor CCR10 which binds the chemokine CCL28 that is constitutively produced by colonic cells [116]. CD4 and CD8 T cells in the LP are found at about the same ratio (2:1, respectively) as those found in the peripheral blood. In contrast, for intraepithelial T cells, the ratio is reversed, with CD8 T cells predominating [117,118,119]. Intraepithelial T cells, as well as those in the LP, typically have an activated phenotype, in contrast to those in the periphery which are ‘resting’ and only activated in response to infection. The unique environment of the gut necessitates that mucosal immunity discriminates between commensal and pathogenic bacteria. Although this discrimination is a complicated process and how it is achieved is not fully understood, it is believed that, in part, the commensals produce SCFAs to stimulate DCs. The DCs then produce transforming growth (TGF)-β and retinoic acid (RA), which polarize CD4 T helper cells toward the Treg subset [120,121]. Conversely, bacteria that adhere to the epithelial layer stimulate DCs to produce cytokines that promote Th17 differentiation (to recruit neutrophils that phagocytose and kill bacteria), thus protecting the host from pathogenic bacteria [122]. 

### 4.2. B Cells and GI Antibody Production

Like T cells, the gut contains the body’s largest populations of B cells, which, in the gut, are mostly activated and terminally differentiated plasmablasts and plasma cells. In fact, the gut contains more than 80% of the body’s activated B cells [123,124]. In addition to antibody production, B cells are also MHC class II-APCs that bind to and endocytose soluble antigens through their B-cell receptor which is a membrane-bound form of immunoglobulin. 

Of the five classes of antibodies, IgA is the most prevalent in mucosal secretions. In contrast to IgA that is found in peripheral blood, plasma cells in the gut mostly produce the dimeric, and to a lesser extent, the trimeric and tetrameric secretory IgA forms (sIgA). sIgA is characterized by a 137-residue polypeptide known as the joining (J)-chain, that couples two (or more) IgAs at the Fc region to form a multimeric immunoglobulin molecule [125,126,127,128]. The J-chain regulates multimer formation but also covalently binds to the polymeric immunoglobin receptor (pIgR) for transport across the epithelium and into the gut lumen [129]. Upon being transported, the pIgR is proteolytically cleaved, releasing the IgA dimer; however, a piece of the pIgR remains covalently bound to the immunoglobin (by disulfide bonds) and is known as the secretory component (SC). The SC is heavily glycosylated with seven N-glycan polymers and is an integral part of the sIgA complex. Likewise, the secreted IgA molecule is highly glycosylated, at each hinge region, as well as with additional N-glycans attached to the J chain. As a result of this extensive glycosylation, the immunoglobin is resistant to luminal proteases [130]. The glycan groups also bind to GI bacteria, and, therefore, provide an important effector function in maintaining microbiota homeostasis, which is independent of the specific binding by the antigen-binding Fab portion of the antibody [131]. The glycosylation of the SC also allows it to interact with GI mucus. Additionally, the SC has biochemical similarities to the epithelium and may compete for binding to intestinal bacteria. IgM produced in the gut also expresses the J-chain, yielding a pentameric form that can be transported into the lumen; however, in contrast to sIgA, the SC bound to IgM is not covalently linked [132].

IgA is essential for maintaining the homeostasis of the microbiota, and perturbations in IgA secretion may lead to dysbiosis (Figure 4). Commensal gut bacteria are typically coated with sIgA which supports immune exclusion to prevent bacteria from translocating across the gut epithelium [133]. Because of the continual loss of sIgA by defecation, IgA antibody is produced in large quantities (>3 gm/day) in healthy individuals [134]. Indeed, at least 80% of the body’s antibody production takes place in the gut [123,124]. Several factors influence this IgA production, including the ligation of TLRs of the IECs by bacterial products and other effects of bacterial products.

Ligation of TLRs of IECs can increase IgA by a series of events that produce RA which promotes B cell production of IgA. TLRs expressed by gut epithelial cells are polarized in their distribution on cells, with TLRs 2&9 primarily on the luminal side of the epithelium and other TLRs on the basolateral side. TLR engagement on the luminal side can downregulate inflammatory signals to increase Tregs [135] or stabilize the inhibitor of NF-κB to keep it from initiating the transcription of inflammatory cytokines [136]. TLR engagement from the basolateral side can generate proinflammatory signaling, through NF-κB activation [136,137]. APCs are also influenced, and they in turn promote the IEC generation of retinaldehyde dehydrogenase (RALDH), the enzyme that produces RA from dietary vitamin A. RA stimulates the migration of naïve B cells to the gut through the upregulation of integrins and chemokine receptors [138,139]. RA also drives class switch recombination of B cells from IgM to IgA production in the GALT [140,141]. SCFAs, including isobutyrate, isovalerate, and 2-methylbuyterate, which are produced by some commensal bacteria, also influence the production of sIgA [142]. Thus, it is not surprising that alterations in the production of sIgA, and the composition of the microbiota are intimately connected [143,144,145]. 

The amount and affinity of sIgA are important for the homeostasis of the gut microbiome [146]. Although individuals with selective IgA deficiency, which is characterized by a severe deficiency or complete lack of IgA with the normal production of IgM, IgG, and IgE, appear relatively healthy, they are more prone to respiratory mucosal-associated infections, as well as allergies and autoimmune diseases [147,148]. This respiratory vulnerability is likely the result of sIgM being absent in the bronchial mucosa but present in the gut lumen where it can perform a compensatory function [149]. 

## 5. Bridges between Innate and Adaptive GI Immune Responses

### 5.1. Conventional Dendritic Cells

DCs are phagocytic cells that reside in tissues and lymphoid organs and are numerous in the gut [150]. Immature DCs migrate through the bloodstream to their target organs or tissue where they develop into two main classifications based upon their cell surface markers and primary functions: cDCs and pDCs [151,152]. cDCs can be further divided based on their surface markers, and the specifics of their primary function, which is to activate T lymphocytes during infections [153,154,155]. This process begins with the intake of microbes by receptor-mediated phagocytosis after binding PRRs, complement receptors, or Fc receptors [156,157]. Additionally, cDCs may incidentally phagocytize microbial proteins through the ingestion of extracellular fluid in the process of macropinocytosis [158]. Antigens from digested microbes are loaded onto and displayed on MHC class II molecules or may be rerouted to the cytoplasm, degraded by the proteasome, and loaded onto MHC class I molecules during cross-presentation to CD8 T cells [153,159]. At the same time, the PRRs used in pathogen uptake, such as the lectin DC-SIGN and a wide range of TLRs, signal the maturation of the DCs mainly through the NF-κB signaling pathway [160,161]. This maturation induces the expression of B7 costimulatory molecules, as well as the CCR7 chemokine receptor, allowing chemotaxis to lymphoid tissue in response to CCL19 and CCL21 [162,163]. The licensed DCs now efficiently prime T cells through their high expression of MHC class I, MHC class II, and B7 costimulatory molecules. The DCs also secrete cytokines such as IL-12 and IL-23 to induce differentiation of naïve T lymphocytes as well as release chemokines to attract naive T lymphocytes to the lymphoid organs to engage in antigen presentation [164,165,166].

In the gut, cDCs reside in PPs and the LP, where they process antigens from the lumen and migrate to MLNs [167,168,169]. In non-inflammatory conditions and in the gut, antigen presentation to naïve T cells results in differentiation of the T cells into FoxP3+ Tregs, which migrate from the MLNs back to the gut via CCR9, RA, and α4:β7 signaling where these T cells induce oral tolerance [170,171] as described in Section 2.4. cDCs are divided into groups, cDC1 and cDC2, based on function as well as the expression of surface markers. cDCs are distinguished by the expression of high levels of CD11c as well as MHC class II. The MHC class II, which is normally limited to only a few cell types, supports the processing and presentation of peptide antigens to CD4 T cells. cDCs also lack the high-affinity Fc receptor CD64, a classical macrophage and monocyte marker [172]. In the gut lymphoid tissue, cDC1s promote Th1 responses and are identified by their expression of CD8a and a lack of CD4 (which is expressed in low amounts on monocytes and macrophages) and CD11b and selectively promote Th1 responses. In contrast, gut lymphoid-associated cDC2s express CD4 and CD11b but lack CD8a and promote Th2, Th17, and Treg responses [173,174]. In the LP, cDC1s can be identified as CD103+/CD11b- and cDC2s are identified as either CD103+/CD11b+ or CD103-/CD11b+ [173,174]. Thus, the phenotypes of the resident DC cells can be informative as to the nature of ongoing immune T-cell responses. 

Like B and T cells, RA also drives cDC trafficking to the gut and impacts transcriptional regulation as well as their development [175,176,177]. However, unlike gut-associated T and B cells, gut-associated cDCs can convert vitamin A into RA through their intrinsic RALDH activity. Consequently, RA produced by resident DCs acts on T- and B-cells and upregulates their gut homing receptors α4:β7-integrin, and CCR9, thus promoting T and B cell homing to the small intestine. Interestingly, previous reports suggest that the microbiota as well as vitamin A in the diet can influence cDC phenotype and function, in part through the production of SCFAs. cDCs that are exposed to butyrate and propionate upregulate RALDH transcription in a process that subsequently promotes FoxP3 expression and Treg differentiation to affect oral tolerance [178]. It should be noted that although cDC1 and cDC2 are present in the periphery as well as the gut, their function and phenotype vary depending on the tissue and their environment. 

### 5.2. Plasmacytoid Dendritic Cells

Although they are bonified professional antigen-presenting cells, the primary function of pDCs involves their prodigious and rapid production of type I and III interferons as well as other signaling molecules, typically in response to viral infections [179,180]. pDCs do present antigens in the context of MHC class II, however, they are much less efficient in doing so when compared to cDCs and only rarely migrate to the lymph nodes from sites of infection [181,182]. It was also reported by Tel et al. that freshly isolated pDCs can cross-present exogenous antigens to CD8+ T cells, although at a substantially lower capacity than cDCs [183]. In addition, pDCs can take up antibody-coupled antigens via their surface C-type lectin DEC-205 receptor, their DC immunoreceptor, and their BDCA-2 (CD303) receptor [184,185,186]. 

Given their consummate role in responding to pathogens, pDCs express several PRRs, including membrane-bound surface receptors, endosomally located receptors, and cytosolic receptors. Their most heavily studied PRRs are the endosomal TLRs-7, and -9 which recognize single-stranded RNA and CpG DNA, respectively, typically of viral origin. Engagement of these TLRs in early endosomes initiates the production of interferon through the activation of the transcription factor IRF-7, which is produced constitutively in pDCs [187]. pDCs also secrete inflammatory cytokines and chemokines such as CCL3, CCL4, CCL5, TNF-α, IL-6, and CXCL8, which are produced in response to late endosomal activation [164,188]. In addition, pDCs can differentiate into cells that appear morphologically like conventional-like DCs if stimulated by microbial infection [189,190]. 

The gut houses a relative abundance of pDCs in the epithelium, PPs, and LP as compared to the rest of the body [191,192]. Their function differs slightly from pDCs in other locations, namely the reduced production of type I interferons due to immunoregulatory conditions in the gut as well as inducing IgA production in the gut through a T cell-independent process [192,193]. Moreover, pDCs play a role in oral tolerance by favoring the differentiation of Tregs while downregulating the differentiation of Th17 lymphocytes along with inducing anergy in specific CD4 T lymphocytes [194,195]. The migration of pDCs to the GI tract is also implicated in the pathology of several diseases. For instance, they are found in increased numbers in the gut of those with inflammatory bowel disease as well as those with human immunodeficiency virus (HIV) infections, where they display an immature, but inflammatory phenotype [196,197,198]. Although the phenotype and function of peripheral pDCs and those residing in the lymph nodes have been well-described [199], those associated with other tissues are less so. In the steady-state, pDCs are rare in most tissues such as the skin, except at sites of infection or inflammation [200,201], but are relatively abundant in the intestine, where they comprise up to 1% of the total cells found in the IE and the LP [192]. 

### 5.3. Macrophages

Macrophages are tissue-resident phagocytes that engulf apoptotic and necrotic cells and cellular debris. They serve a much broader role by bridging the innate and adaptive immune system through cytokine production and antigen presentation [202]. While macrophages were once thought to develop from bone marrow-derived monocytes which trafficked into target tissues, it is now recognized that macrophages are derived from a variety of sources, with many populations being established before birth and maintained independently of circulating monocytes [203]. Macrophages are typically divided into two classifications, M1 and M2 (also known as classically activated and alternatively activated, respectively), based on their functions and cytokine secretions. Macrophages exposed to Th1 cytokines such as IFN-γ and GM-CSF, as well as bacterial LPS, are polarized towards the M1 phenotype, which secrete pro-inflammatory cytokines including IL-1β, IL-6, IL-12, IL-23, and TNF-α [204]. M2 macrophages are polarized in response to Th2 cytokines, primarily IL-4 and IL-13, are involved in tissue repair, wound healing, and immunosuppression, and secrete anti-inflammatory cytokines such as IL-10 and TGF-β [205]. M1 and M2 represent extremes on a phenotypic continuum, with each classification representing a spectrum of heterogeneous activation states [206].

Given their proximity to the billions of bacteria in the GI tract and the importance of clearing any bacteria that breach the single-cell layer of the gut epithelium, it is not surprising that the largest population of resident macrophages in the body is located in the GI mucosa [207]. While many macrophages are found in the subepithelial area in the LP, others are found in deeper layers, such as the submucosa and muscularis [208]. During neonatal development, the gut is populated by yolk-sac-derived macrophages, most of which are replaced by bone marrow-derived macrophages after birth [209]. However, self-maintaining populations of embryonic macrophages persist and are located near blood vessels, enteric neurons, Paneth cells, and PPs, and appear to regulate intestinal permeability, vascular function, and intestinal motility [210]. Macrophages in the gut can be distinguished from those elsewhere in the body by their surface markers. Although much of our understanding comes from work with mice, several surface markers can be used across species. For instance, the high-affinity Fc receptor CD64 and the LPS-binding protein CD14 are present in humans and murine macrophages, as are CD4, CD163, CD172a, and CD206 [211,212,213,214,215]. Notwithstanding, the expression level of these surface markers may differ depending on their location in the gut [211]. 

Like macrophages found elsewhere in the body, those in the gut wall are prodigiously phagocytic. Unlike macrophages found in other tissues, gut macrophages do not necessarily promote overt inflammation upon phagocytosis of bacteria but can promote tolerance, preventing inflammation from constant exposure to bacterial antigens [211,216,217]. Intestinal macrophages are characterized by the secretion of high levels of immunosuppressive IL-10, which likely contributes to decreased responsiveness to TLR-activating stimuli, such as their constant exposure to LPS [218].

Studies in mice indicate that subepithelial macrophages extend their transepithelial dendrites between the epithelial layer and “sample” the environment of the lumen [219]. This process does not compromise the epithelium nor the tight junctions that form the selective seals between the epithelial cells; it likely promotes tolerance to commensal bacteria given that it is enhanced by pyruvic acid and lactic acid, two common microbial metabolites [220]. Specific macrophage populations also use this mechanism to sample apoptotic IECs [221]. Although it has been proposed that tolerance to orally consumed antigens may occur through this sampling mechanism, as mentioned previously, oral tolerance is initiated in the MLN, and these macrophages do not migrate to the MLN under normal conditions [220,222]. Moreover, naïve T cells are generally absent in the LP where these macrophages are [220,222]. Nonetheless, macrophages can transfer captured antigens to migratory DCs for presentation to T cells in the MLN [223,224], suggesting a cooperative role is possible to influence the MLNs. Furthermore, through macrophage production of anti-inflammatory cytokines, such as IL-10 and TGF-β, they facilitate the secondary expansion of Tregs locally in the LP, so they likely play an important, part in the induction of oral tolerance.

Macrophages may directly affect the gut–brain axis. In the muscularis, they may regulate enteric neurons, impacting GI smooth muscle contractions and, in turn, peristalsis [225]. Macrophage-derived bone morphogenic protein (BMP) 2 activates the BMP receptor on enteric neurons, which leads to the secretion of colony-stimulating factor 1, a growth factor required for monocyte/macrophage development in the bone marrow. Intriguingly, when broad-spectrum antibiotics are administered, BMP-2 production and peristalsis are ablated suggesting that microbiota-driven crosstalk between muscularis macrophages and enteric neurons controls GI motility. Macrophages also receive signals via the vagal nerves, which negatively regulate their cytokine production in response to acetylcholine activation of nicotinic acetylcholine receptors (nAChR) [226]. Moreover, nAChR activation of macrophages promotes the surveillance of luminal antigen through macrophage phagocytosis, as well as by inducing a temporary increase in epithelial permeability [226]. Thus, initial disturbances of either gut resident macrophages or neurons may be associated with effects on the other cells.

### 5.4. Natural Killer Cells

Natural killer cells (NK) are bone marrow-derived cytotoxic lymphocytes that belong to the ILC family, and primarily target pathogen-infected and malignant cells. Potential targets for NK cell killing are largely determined based on their surface expression of activating and inhibiting proteins. Activating proteins are typically upregulated because of cellular stress, such as the direct viral infection of GI epithelial, whereas inhibitory proteins are those typically expressed in large amounts on healthy cells but can be downregulated when a cell is malignant or in response to evasive capacities of different viruses [227,228]. One such example is MHC class I, which is constitutively expressed on all nucleated cells but is downregulated during certain viral infections, as the cellular machinery is redirected to making viral proteins or the virus affects MHC class I functions [229]. The ratio between activating and inhibiting signals impacts whether the cell will be targeted for killing [230]. 

Analogous to cytotoxic CD8 T cells, NK cells can induce apoptosis through either extrinsic or intrinsic pathways [231]. To support extrinsic cytotoxicity, NK cells are replete with cytotoxic granules, containing the enzymes granzymes and the pore-forming protein perforin. Upon binding to a target cell, an immunological synapse is formed, which polarizes the cytotoxic contents of the cell toward the target. This synapse results in the focused delivery of granzymes and perforin to the target cell, which then induces death that can occur with or without activation of procaspases [232,233]. Alternatively, NK cells may trigger apoptosis through the intrinsic pathway, via their TNF family receptors TRAIL and FasL, which bind to death-inducing ligands such as Fas, DR4, or DR5 expressed on target cells [234]. The engagement of these ligands triggers apoptosis by activating the zymogen procaspase 8 [235,236]. NK cell killing efficiency can be increased in response to cytokines and interferons. Death can also be directed by the binding of antibodies on target cells to the NK cell Fc receptors in the process of antibody-dependent cell-mediated toxicity (ADCC) [237,238]. Lastly, NK cells become activated after interacting with infected cells to subsequently produce and secrete inflammatory cytokines such as TNF-α, and antiviral cytokines such as IFN-γ to confer further viral protection [239,240]. 

Circulating NK cells and those located in the enteric system have differing features, such as their receptors, what they secrete, and their levels of cytotoxicity [241]. For instance, NK cells express the neural adhesion marker NCAM, also known as CD56. The level of CD56 expression, bright vs. dim, is associated with different functional roles. Approximately 90% of circulating NK cells are CD56^Dim^ CD16^+^ whereas the other ≈10% are CD56^Bright^ CD16^−^ [242]. In the periphery, CD56^Bright^ CD16^−^ NK cells predominate in lymph nodes and sites of inflammation and when activated produce copious amounts of cytokines, such as IFN-γ, TNF-α, GM-CSF, IL-10, IL-5, and IL-13 [227]. NK cells may serve different immunoregulatory functions, for instance, through the expression of the high-affinity IL-2 receptor, NK cells may compete for IL-2 with other cells [243]. The Fc receptor CD16a (FcγRIIIa), which is expressed on CD56^Dim^ NK cells binds the constant portion of IgG, activating ADCC. Not surprisingly, CD16a is not expressed to any significant extent on gut-associated NKs where the predominant antibody isotype is IgA, which cannot engage this Fc receptor [244]. Therefore, gut NK cells likely play a greater role than their circulating counterparts with respect to maintaining gut homeostasis through cytokine production. In fact, NK cell-produced IL-22 plays an important role in maintaining gut epithelial cell survival [245]. 

## 6. Human Neuroimmune Diseases with Altered Microbiomes

Several neuroimmune diseases have known associations with altered microbiotas, potentially implicating the gut-microbiota-brain axis in their pathophysiology. In the following section, we highlight some commonly known neuroimmune diseases with such associations as examples of the potential impact of mucosal immunity on these diseases. However, several excellent reviews are available that provide a more comprehensive analysis of each respective disease [60,246,247].

### 6.1. Alzheimer’s Disease (AD)

Pathologically, AD is associated with a toxic buildup of Aβ amyloid plaques and hyperphosphorylated and misfolded tau protein that initiates in the hippocampus and eventually spreads to the cortex [248]. Recent evidence suggests that alterations in the microbiome of AD cases may contribute to the pathophysiology of AD [249]. Supporting a microbiome involvement in AD pathophysiology, Minter et al. reported that antibiotic treatment of a murine model of AD leads to reduced amyloidosis [250]. Kobayashi et al. reported that oral administration of *Bifidobacterium breve* strain A1 to a mouse model of AD (induced by administration of Aβ25–35 into the cerebral ventricles) reversed cognitive impairment [251]. Wang et al. showed that, in contrast to wild-type mice, the gut microbiota in AD mice (5xFAD model) spontaneously changes over time and facilitates brain infiltration of immune cells, leading to microglial activation, cognitive impairment, and Aβ amyloidosis [252]. These important studies strongly support that an altered microbiome contributes to the neuropathological progression of AD. Importantly, Kobayashi and colleagues transcriptional profiled the hippocampus and identified 305 genes that were differentially expressed in the AD animal model when compared to non-AD mice, and most of these genes were involved in the immune response. Strikingly, upon treatment with *B. breve* A1, the transcriptional profiles of AD mice differed from non-AD controls by only two genes, suggesting that *B. breve* A1 could regulate excessive AD-associated immune responses. These experiments underscore the potential role of the microbiome in AD pathology. 

An increasing number of studies also suggest that the microbiota and AD pathophysiology are interconnected. For example, Vogt et al. showed that the gut microbiome of AD cases contained less microbial diversity and was compositionally distinct from matched non-AD controls [253]. Recent evidence shows that a significant fraction of brain-derived neurotrophic factor (BDNF) is produced in response to the GI microbiota [254]. Considering the neuroprotective role of BDNF, it would be relevant to establish if an altered microbiota reduces BDNF, and thereby, exacerbates AD neuropathology. Notwithstanding, a reduction in the level of BDNF and neuroprotective signaling is observed in postmortem brain tissue as well as in vivo models of AD. Loss of BDNF has been suggested to contribute to the overt and progressive neurodegeneration of hippocampal neurons [255]. A reduction in BDNF may also exacerbate oxidative stress and alter gut homeostasis in AD. These studies make a strong case for the involvement of the gut-microbiota-brain axis in AD. 

### 6.2. Autism Spectrum Disorders

Autism spectrum disorders (ASD) are neurodevelopmental disorders characterized by deficits in social interactions and communication, repetitive and stereotyped behaviors, anxiety, and cognitive disturbances. Many children with ASD report associated GI symptoms, the most frequent are chronic diarrhea, gaseousness, and abdominal discomfort and distention [256]. A meta-analysis study investigating GI symptoms among children with ASD surveyed 15 studies (1980–2012) and found that ASD children experience significantly more general GI symptoms than comparison groups, with higher rates of diarrhea (OR, 3.63; 95% CI, 1.82–7.23), constipation (OR, 3.86; 95% CI, 2.23–6.71), and abdominal pain (OR, 2.45; 95% CI, 1.19–5.07) [257]. Histologic examination of 36 children with autistic disorder revealed grade I or II reflux esophagitis in 25 cases (69.4%), chronic gastritis in 15 cases, and chronic duodenitis in 24 cases. The number of Paneth’s cells in the duodenal crypts was significantly elevated in autistic children compared with non-autistic control subjects. Low intestinal carbohydrate digestive enzyme activity was reported in 21 cases (58.3%), although there was no abnormality found in pancreatic function [256].

Recent studies suggested that GI microbiota may play an important role in ASD. The possible mechanisms include increased intestinal permeability “leaky gut” [258,259], overall microbiota alterations [260,261], or gut infection by *Clostridium* spp. [262,263]. Increased intestinal permeability was found among patients with autism (36.7%) and their relatives (21.2%) compared with control subjects (4.8%) [264]. Moreover, children with ASD are very selective “picky” eaters, and most of them show aversions to specific food colors, textures, smells, or other food characteristics [264]. Subjects on a gluten-casein-free diet had significantly lower intestinal permeability compared with autistic cases who were on an unrestricted diet as well as healthy controls [259]. As a result of increased permeability, toxins and bacterial products can potentially enter the bloodstream, ultimately affecting brain function and impairing social behavioral scores [265]. The leaky gut also increases the antigenic load from the GI tract. Thus, lymphocytes and ASD-associated cytokines, like IL-1β, IL-6, IFN-γ, and TNF-α, circulate and cross the blood–brain barrier. IL-1β and TNF-α are responsible for inducing immune responses in the brain by binding to the brain endothelial cells [67,265,266].

Significant alterations in microbiota composition and metabolites occur in children with ASD. Children with autism have lower levels of *Bifidobacter* spp. and higher levels of *Lactobacillus* spp., but similar levels of other bacteria and yeast using standard culture growth-based techniques [267]. De Angelis et al. found that *Caloramator*, *Sarcina*, and *Clostridium* genera were high in ASD children when compared to healthy children [260]. The composition of the Lachnospiraceae family also differed in ASD children. Except for *Eubacterium siraeum*, a low level of Eubacteriaceae was found in fecal samples of ASD children [260]. Finegold et al. examined the fecal microbial flora of 33 subjects with various severities of autism with GI symptoms, seven siblings not showing autistic symptoms (sibling controls), and eight non-sibling control subjects. Bacteroidetes were found at high levels in the severely autistic group, while Firmicutes were more predominant in the control group. Smaller, but more significant, differences also occurred in the Actinobacterium and Proteobacterium phyla. *Desulfovibrio* sp. and *Bacteroides vulgatus* were present in higher numbers in the stool of severely autistic children when compared to controls [261]. A recent randomized, double-blind, placebo-controlled study demonstrated that a combination of a casein/gluten-free diet along with the Bimuno^®^ galactooligosaccharide (B-GOS^®^) prebiotic led to an improvement in the behavioral symptoms of autistic children [268].

In terms of Clostridia, both the gastric and small-bowel specimens from children with autism were more likely to have Clostridia and more likely to have a higher number of species of Clostridia than control specimens [263]. ASD children treated with oral vancomycin have improved in behavioral, cognitive, and GI symptoms [269]. Because vancomycin is only minimally absorbed when given orally, it is likely that the effect is mediated through its activity on intestinal bacteria. The relapse after discontinuation of therapy may be related to the persistence of spores from spore-forming organisms such as Clostridium that germinate after the drug is stopped [263].

### 6.3. Parkinson’s Disease 

Parkinson’s disease (PD) is a relentless, chronic, neurodegenerative disorder that is characterized by the progressive loss of axons from dopamine neurons. When approximately 90% or more of midbrain dopamine neurons are lost, PD patients experience tremors in one or two limbs (upper or lower), corporal instability as evidenced by a loss of balance, and shuffling gait [270]. PD is an age-related disease that affects the elderly; the incidence of PD increases after 60 years of age and exponentially in individuals aged 80 years and older [271]. 

Beyond motor symptoms, the onset of non-motor symptoms is common in PD including cognitive decline, altered sleep patterns, vagal nerve dysfunction leading to alterations in blood pressure regulation, and robust GI dysfunction [272,273,274,275]. These non-motor symptoms can present 10–15 years before the onset of motor symptoms in PD patients. For instance, constipation is a common non-motor symptom and a plausible diagnostic criterion for early-stage disease [276]. Over 50% of those with PD suffer from severe constipation and are comorbid for Crohn’s disease and inflammatory bowel disease (IBD) raising the possibility that the integrity of mucosal barriers and GI epithelial tissue are severely compromised [277]. This damage can contribute to severe inflammation and GI dysfunction. Consistent with this pathological model of PD, one clinical study found evidence of leaky gut indicated by: widespread inflammation and disrupted GI epithelial tissue; an increase in calprotectin as well as zonulin, two well-validated biochemical markers of gut inflammation and oxidative stress, in fecal matter and serum in more than 50% of PD cases [278]. Additionally, immunohistochemical data suggest that the integrity of the gut barrier is compromised in PD cases as evidenced by alterations in tight junctions in GI epithelia [279]. In PD cases, the expression of claudin, occludin, and occludens-1—three structural proteins that are involved in the assembly and maintenance of tight junctions—is aberrantly altered in the colon in PD cases and is associated with increased permeability of intestinal epithelial barriers [65,280]. 

The disruption of barrier integrity occurs concomitantly with a significant increase of inflammatory markers including TNF and several other pro-inflammatory cytokines in PD cases compared to age-matched controlled cases. Interestingly, it is worth noting that these inflammatory cytokines are increased in PD with a somewhat similar cytokine profile as observed in IBD. Additionally, the disruption of barriers was positively correlated with the onset of IBD in PD [278]. Linking a PD-associated protein with gut pathology, α-synuclein, a cytoskeletal protein that forms intracellular aggregates in the brain termed in Lewy bodies, –accumulates in enteric nervous system (ENS) neurons [281] and gut epithelium. These aggregates may contribute to the progressive degeneration of gut tissue [282].

While the pathological mechanisms that disrupt the integrity of GI tissue remain to be elucidated, there is clear evidence that gut microbiome dysbiosis occurs in PD and may generate oxidative stress and chronic inflammation there. Indeed, a variety of bacterial families are altered in PD, including Lactobacillaceae, Bacteroides, Prevotella, Clostridium, and many other bacterial families [283,284,285]. Additionally, recent experimental in vivo evidence suggests that microbiome dysbiosis in PD causes an accumulation of α-synuclein in the gut. For instance, fecal transplantation studies show that transplanting microbiota from PD humans worsened the motor symptoms in α-synuclein transgenic mice and worsened the protein aggregation, whereas antibiotic treatment ameliorated motor symptoms [286]. 

While the molecular etiology that contributes to gut pathology is largely unknown, recent in vivo evidence suggests that gut inflammation may be driven via the early activation of innate immune cells, including pDCs that may migrate to the gut epithelium. One study found that the number of mature cDCs and pDCs in the bloodstream of PD patients significantly decreased during motor symptom severity [287]. While clinical studies that analyze the recruitment of cDCs or pDCs in the GI of PD cases are yet to be conducted, the cited study suggests that pDCs may be lower in blood because they are being recruited to diseased sites, including the brain and the GI tract. A direct link between several signaling proteins associated with PD and gut homeostasis has raised the possibility that signaling pathways are disrupted in PD and thereby contribute to GI pathology. Recent papers show that at the molecular level, Leucine-Rich Repeat Kinase 2 (LRRK2), an atypical serine/threonine kinase that is mutated in the autosomal dominant form of familial PD, is directly involved in the regulation of immune system cells. LRKK2 activates inflammasomes to facilitate macrophage- and neutrophil-mediated clearance of *Salmonella typhimurium* GI infections [288]. mRNA profiling indicates that constitutive expression of LRRK2 is high in peripheral blood mononuclear cells (PBMCs), even to a greater extent than in neurons. LRRK2 is highly expressed in primary monocytes and macrophages can be induced in activated T cells and is present during the differentiation of naive DCs to mature DCs. Overall, alterations in this kinase may be a real trigger for PD symptoms.

Other immune-modulatory roles of LRKK2 include eliciting IFN-γ responses [289], facilitating phagocytosis of bacteria by macrophages, and regulating immune signaling pathways involved in the pathogenesis of Crohn’s disease. LRKK2 is induced in inflamed colonic tissue in Crohn’s disease patients, suggesting that this kinase may mediate pathogenesis in the gut [289]. However, while these observations suggest that LRRK2 modulates immune functions in the gut, a direct link of LRRK2 with GI pathology in PD is yet to be established.

A second PD-associated protein interacts with LRRK2 in a linear signaling pathway and can act synergistically in PD pathogenesis. This protein, α-synuclein, has been recently shown to modulate gut immune functions [290]. α-Synuclein is a presynaptic protein that is prone to aggregate and will form Lewy bodies in the brain and gut tissue in PD. α-synuclein is a chemoattractant to elicit the migration of neutrophils and monocytes in gut tissue as measured in vitro [291]. Interestingly, α-synuclein can stimulate the maturation of DCs and enhance the production of inflammasome-associated cytokines IL-1β and IL-6 [291,292]. In aggregate, these data suggest that α-synuclein, which progressively accumulates in the ENS and GI epithelium, may stimulate immune cell populations to migrate to the gut epithelium and foment cell-mediated immunity in response to microbiome alterations during PD progression. Since LRRK2 and α-synuclein are both induced in inflamed GI epithelium, and regulate GI immunity [293,294], it is conceivable that LRRK2 interacts directly with α-synuclein to increase cytokine production. However, given that both LRRK2 and aggregated α-synuclein regulate the function of pDCs [295], it is not clear whether activated pDCs in the gut further contribute to gut inflammation, through the production of inflammatory cytokines, or disrupt gut epithelial barrier integrity. To this end, it would be beneficial if future in vivo studies that test these two hypotheses are performed to further understand the molecular mechanisms that drive early GI pathology in PD.

### 6.4. Myalgic Encephalomyelitis

Myalgic encephalomyelitis (ME) is a disease characterized primarily by debilitating fatigue complemented by a variety of other symptoms, including musculoskeletal pain, neurological impairment, sleep disturbances, and GI dysfunction [296,297]. ME is accompanied by mitochondrial dysfunction, increased oxidative and nitrosative stress, and profound immune dysregulation [296,298,299,300,301]. A variety of autoantibodies are often observed in ME cases, including multiple antinuclear antibodies and antibodies against ß-adrenergic and muscarinic acetylcholine receptors [302,303,304,305,306]. These antibodies have been linked to altered brain network structure and hypothesized to explain symptoms of ME through altered vascular and neural function [307,308]. 

Disruptions to the intestinal epithelium may be common to both ME and inflammatory bowel diseases, involving both altered immunity and microbiomes [309]. Increases in serum IgA and IgM against the LPS of Enterobacteria, as well as an increase of *Bacteroides* sp. in the gut microbiota, indicate that increased gut permeability likely plays a role in the pathophysiology of ME [310,311]. Giloteaux et al. reported that for ME there is an increase in plasma LPS, liposaccharide binding protein (LBP), and the LPS/LBP receptor sCD14 [312], all of which are biomarkers of microbial translocation [313].

Metagenomics studies using 16S rRNA sequencing of the gut microbiome have confirmed that ME is accompanied by an alteration in microbiome composition. Giloteaux et al. found an overall reduction in microbial diversity with decreased abundance of Ruminococcaceae and other Firmicutes and a two-fold increase in Enterobacteriaceae abundance [312]. Lupo et al. reported a reduction in Lachnospiraceae and increases in Bacteroides and Phascolarctobacterium [310]. Using biochemical identification tests, Sheedy et al. found an increase in aerobic Gram-positive bacteria, especially Streptococcus and Enterococcus, which they hypothesized to increase D-lactic acid production and intestinal permeability [310]. Lastly, Fremont et al. reported that microbiomes differ between ME cases from controls and can also differ among distinct geographic locations [314]. 

Mucosal-associated invariant T (MAIT) cells are innate-like T cells that express semi-invariant α/ß T cell receptors [315]. They are restricted by MHC class I-like antigen-presenting molecule 1 (MR1) and detect bacterially infected cells through the presentation of intermediates of microbial riboflavin biosynthesis that are bound to the MR1 proteins [316,317]. MAIT cells can also be activated in an MR1-independent manner through the binding of the cytokines IL-12 and/or IL-18 [318]. MAIT cells are activated in inflammatory bowel diseases, with Serriari et al. reporting increased frequency and activation of MAIT cells in inflamed mucosa in subjects with Crohn’s disease and ulcerative colitis [319]. Cliff et al. reported increased proportions of MAIT cells in subjects with ME and found that the MAIT cells in those with severe ME were heavily skewed towards the CD8+ subset [320].

Human endogenous retroviruses (HERVs) are genomic elements associated with ancient retrovirus infections. Studies have suggested that HERVs may play a role in neuroinflammatory diseases such as multiple sclerosis and amyotrophic lateral sclerosis [321]. De Meirleir et al. reported immunoreactivity to HERV proteins Gag and Env in a subset of ME cases with GI comorbidity, and further identified that this immunoreactivity was uniquely found in pDCs. They also reported duodenum-associated pDC concentrations in ME cases approximately 4.7 times as high as in controls, suggesting pDC involvement in the GI pathology of ME [322]. Previous studies have reported that the expression of HERV –H, -W, and -K is significantly impacted by the presence of specific microbes in the gut [323]. Interesting Yu et al. reported that TLR-7 and 9 inhibition promotes the expression of HERV sequences specifically in pDCs [324], suggesting a possible connection between the gut microbiota and HERV expression, potentially through the inhibition of TLRs; however, further studies are needed to confirm this possibility.

### 6.5. Multiple Sclerosis

Multiple sclerosis (MS) is a devastating autoimmune disease, identified by chronic inflammation of the CNS, leading to demyelination. Although the etiology of MS is presently unknown, genetics and environmental factors are believed to play a key role [325,326]. In addition to numerous neurological symptoms, those with MS commonly present with GI abnormalities [327]. Indeed, a survey-based study revealed that approximately two-thirds of those with MS reported GI issues that persist for at least six months, which include, constipation, diarrhea, and fecal incontinence [328]. 

Previous studies have reported that subjects with MS have an altered microbiome when compared to matched controls [329,330,331]. For example, Miyake et al. conducted a longitudinal study in Japan and compare the gut microbiota of subjects with relapsing-remitting MS (RRMS) to that of healthy controls and reported that 21 species of bacteria showed significant alterations in the relative abundance as well as observed an overall moderate dysbiosis in the RRMS cohort. Conversely, in contrast to other diseases, such as inflammatory bowel disorders, which show reduced diversity [332,333], they reported that RRMS cases displayed similar bacterial diversity to that of controls. 

Perhaps the most convincing evidence for a microbiota-MS association stems from observations made using the experimental autoimmune encephalomyelitis (EAE) mouse model of MS. Lee et al. observed that intestinal microbiota significantly influences the balance between proinflammatory and anti-inflammatory and immune responses during the induction of EAE [334]. Specifically, they reported that mice, reared under germ-free conditions developed an attenuated form of EAE characterized by decreased levels of the proinflammatory cytokines IL-17A and IFN-γ in the intestine and spinal cord with an associated increase in Tregs. They additionally showed that specific pathogen-free mice that harbor segmented filamentous bacteria fully developed EAE, thus providing convincing evidence that the bacterial composition of the gut can influence neurologic inflammation in MS. 

In a subsequent study, Haghikia and coworkers showed that long-chain fatty acids (LCFAs) promote polarization of naive T cells toward a Th1 and Th17 differentiation and impaired their intestinal sequestration via the p38-MAPK signaling pathway. In contrast, EAE mice treated with SCFAs exhibited increased differentiation and proliferation of Tregs and an incidental resolution of EAE pathology. It is notable that microbiome survey studies of other neuroimmune diseases such as ME [312] and autoimmune diseases, such as Crohn’s disease [335], are characterized by reduced levels of butyrate-producing bacteria. 

### 6.6. Hunting’s Disease

Huntington’s disease (HD) is a progressive neurodegenerative disease that presents with a specific phenotype of chorea and dystonia, incoordination, cognitive decline, and neuropsychiatric symptoms [336]. Genetically, HD is inherited in an autosomal-dominant fashion with the onset of symptoms typically occurring in the third or fourth decade of life; however, symptoms can manifest any time after infancy (reviewed in [337]). Recent studies in animal models suggest that an altered microbiota may be associated with HD, thus implicating the gut-microbiota-brain axis in the pathophysiology of HD progression. Using the R6/1 transgenic mouse model of HD, Kong et al. reported that male HD mice displayed significant differences in the composition of their microbiome when compared to wild-type mice, although the same difference was not observed for the female mice. However, at 12 weeks of age, dysbiosis was observed which was further associated with impairment in body weight as well as motor deficits [338]. In a subsequent study, Guburt et al. reported that HD gut dysbiosis and cognitive symptoms were ameliorated in R6/1 transgenic mice that received a fecal transplant from wild-type mice [339]. Interestingly, these observations were more pronounced in female mice than in males, potentially reflective of the greater dysbiosis observed in the male mice [338]. Importantly, the fecal transplant was not stably maintained in the transgenic mice, suggesting that the genetic defect associated with the mutant HD gene may impact mucosal immunity; however, further studies will need to be conducted to address this possibility. 

Work conducted by Du and colleagues confirmed that the microbiomes of human subjects with HD were also different from age and sex-matched controls, consistent with the results of Kong et al. in transgenic mice, with HD cases showing increased alpha (richness) and beta (structure) diversity [340]. However, in contrast to the observations made using the R6/1 transgenic HD model, differences between cases and controls were not dependent on the gender of the subject. To identify potential immunological differences that associated with clinical presentation, plasma cytokines were also assessed. Circulating IL-4 was showed to be decreased in HD cases (*p* = 0.03) and correlations between components of fecal microbiota and cytokines were identified between Intestinimonas and plasma IL-4 (*p* = 0.028), and Bilophila and plasma IL-6 levels (*p* = 0.001). However, establishing a cause-and-effect relationship was beyond the scope of the study. Although the work to understand the contributions of the microbiota to the pathophysiology of HD is in its infancy, these important studies [339,341,342], as well as others, have identified a promising area of investigation, whereby modulation of the gut microbiota may provide therapeutic opportunities in treating this devastating disease. 

## 7. Conclusions

While the gut microbiota and its connection to neuroimmune disease have become one of the most active areas of neurological research, the underpinnings of mucosal immunity that impact the microbiota have received little attention. In this review, we have endeavored to bring to light the contributions of mucosal immunity that may impact the gut microbiota and, have used as examples, six neuroimmune diseases where alterations in mucosal immunity are manifested. It should be noted that whether changes in mucosal immunity led to changes in the gut microbiota or the altered microbiota perturbed mucosal immunity is largely unresolved in these diseases and could differ among the six diseases that were highlighted. 

## Figures and Tables

**Figure 1 ijms-23-13328-f001:**
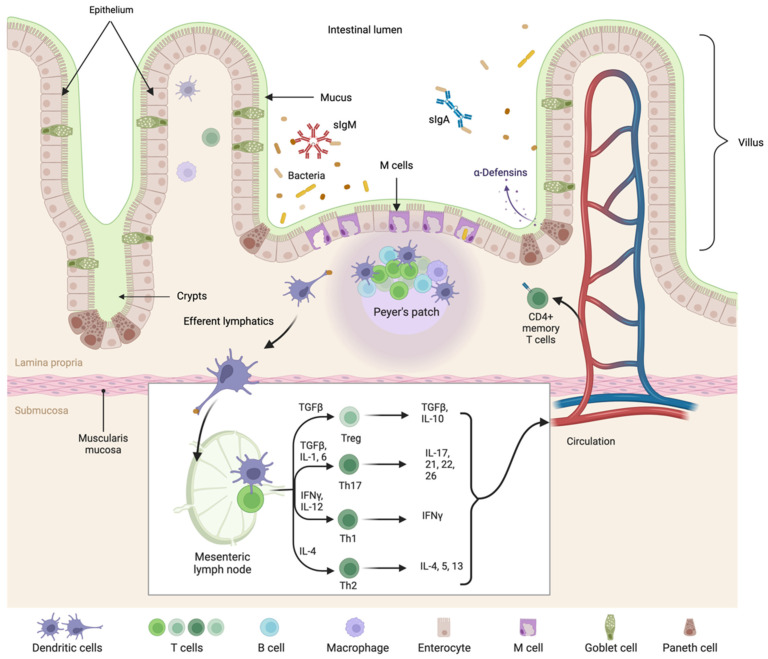
The architecture of the small intestinal GALT [20].

**Figure 3 ijms-23-13328-f003:**
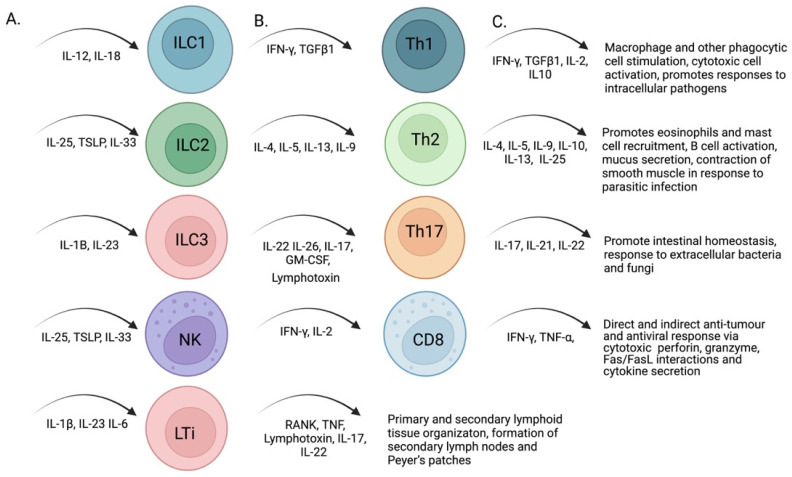
Actions of cytokines produced by innate lymphoid cells. Innate lymphoid cells respond to cytokine engagement (**A**) by producing additional cytokines that act directly on T cells (**B**), to promote additional cytokine production as well as influence their effector function (**C**) [105].

**Figure 4 ijms-23-13328-f004:**
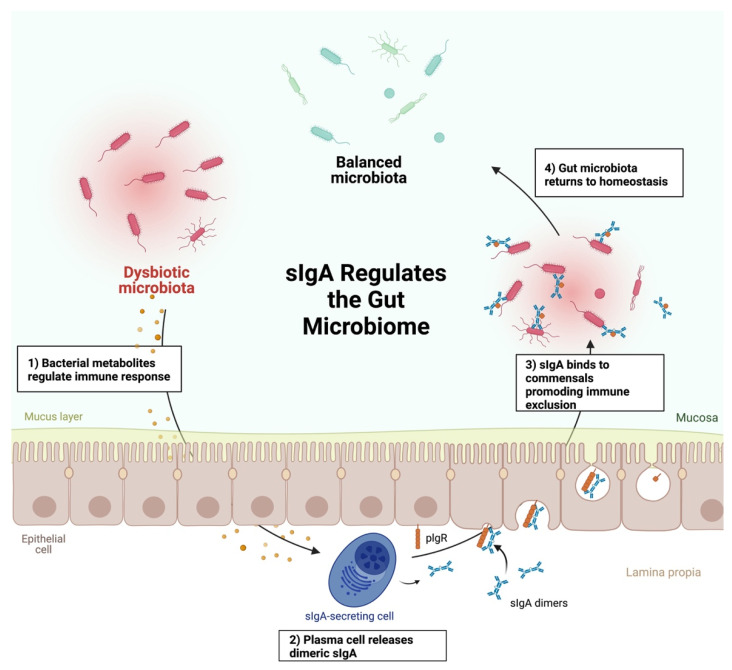
Secretory IgA promotes GI homeostasis through immune exclusion. Dysbioticbacterial metabolites promote cell-mediated immune responses (1). Activation of these responses promotes the production of dimeric IgA by plasma cells (2). Dimeric IgA covalently binds to the polymeric immunoglobin receptor (pIgR) and is transported from the basolateral side of the gut epithelium to the apical side, where it is released by luminal proteases and subsequently binds to commensal bacteria (3), promoting GI homeostasis (4) [105].

## Data Availability

Not applicable.

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
