# Peer review of "Mucosal Immunity and the Gut-Microbiota-Brain-Axis in Neuroimmune Disease"

_ijms, 2022, doi:10.3390/ijms232113328_

Round 1

Reviewer 1 Report

The authors present a review article on a very interesting topic wherein the various constituents of mucosal immunity which may potentially impact the gut microbiota is discussed in detail. Authors have also given specific examples relating to few neuroimmune diseases wherein the modulations in immunological conditions are shown to confer a prominent effect on mucosal immunity. The topic chosen for the article is clinically relevant and it has been not particularly referred to in a lot of scientific articles. This also consequently shows a probable interest in the scientific reader community.

General comments:
Many relevant scientific details have been discussed in detail however the manuscript shows an immense lack of organization and structure. Many of the introductory details presented by the authors have been already discussed in several research articles and the redundancy should have been avoided. Although the authors have attempted to demonstrate the relation between mucosal immunity and the gut-microbiota-brain axis, many of the subsections seem to drift away from the main focus of the article. This might cause the readers to get distracted in the course of reading and therefore it needs to be substantially changed. A review article should generally be very focused and should highlight novel findings relating to the main topic, rather than describing the already existing and well-known established facts in detail. A specific suggestion would be to concise the common facts relating to the subtopics and focus on more associated disease areas, which would also enhance the readability of the article.

There needs to be scientific illustrations, high quality reproduced images from the published articles (with appropriate copyright permissions) and tables which can help in providing a concise version of certain sections. However, the current manuscript does not show adequate figures or tables. Therefore, the current version needs to be significantly revised.

Specific comments:
Few points of consideration in addition to the general comments mentioned above are given here:

1.       Abstract section, second last sentence – “the goal of this review is to…..gut microbiota”- this sentence might not be accurate for abstract and needs to be either eliminated/ rephrased.

2.       Section 2- architecture of the gastrointestinal immune system – this section as indicated above shows a very high redundancy of details which have already been very well established and published in several articles. It is highly recommended to revise this entire section and provide the high level details with concise subsections.

3.       The resolution of figure 1 is very poor. Similarly figures 2 and 3 also lacks resolution. Copyright permissions need to be sought if these are reproduced images and should be indicated appropriately.

4.       Section 3.2, paragraph 2-last sentence (line number 264-265)-“PRRs can occur in many neuronal sites”- please provide adequate literature support by citing reference articles.

5.       Section 3.4- one suggestion is to consider transforming the details given in this section such as subtypes of ILCs to a table. This helps to improve the ease in understanding and also makes the section succinct. Similarly, please consider usage of tables wherever appropriate.

6.       Section 5 is relevant to the context of article. However, this section also seems to be very long with details which have been already established. The authors also have not shown any relevant figures relating to this section. Again, the usage of illustrations and tables need to be exploited as this is a review article.

7.       Section 6.1 on AD- lines 688-689- there is a single bracket that is open which hasn’t been closed. Please clarify.

8.       What is the rationale behind choosing these four neuroimmune diseases specifically? How about similar evidence for other neuroimmune diseases?

9.       Section 6- please consider giving figures of important results described in the subsections which can be reproduced figures from the cited papers (with permission).

10.   Section 6.3- first paragraph on PD- as indicated before, these details are well established and need not be explained again. Authors can describe PD in 1-2 brief statements and then get into the details relating to the context of article.

11.   Section 6.3- paragraph 2- many statements lack reference citations. Examples being lines 791, 801 etc. Similar case for line 819 in paragraph 4.

12.   Section 6.3, page 18, last paragraph- lines 862-872- please indicate appropriate reference articles.

13.   Section 7- conclusions- line 933- ‘manifest’ should be ‘manifested’.

14.   There are several articles describing gut dysbiosis and related alterations causing the pathogenesis of Huntington’s disease (doi: 10.1093/braincomms/fcaa110, doi.org/10.3389/fimmu.2020.603594, doi.org/10.3389/fnins.2022.90220, doi.org/10.1016/j.isci.2021.103687 etc.). Authors should also consider adding this relevant section.

15.   Similarly, multiple sclerosis is another disease area wherein there have been established implications of gut microbiota (https://doi.org/10.3389/fimmu.2022.785644). Please consider adding similar relevant and associated sections so that the major disease areas are covered.

16.   There was not much evidence on microbial–host metabolism interactions in the context of neurodegenerative disorders. A relevant and recent 2022 paper by Gubert et al. https://doi.org/10.1093/gastro/goac017 can be included.

Author Response

We have recited the reviewer’s comments below and have indicated the corrective actions below the corresponding comment below as underlined text. We have also used track changes in the original word document so to make it easier to follow.

We thank the reviewer for their articulated and insightful comments. We appreciate the time and effort it took to make these suggestions and believe your suggestions have made the manuscript much better and more readable.

Thank you,

Vincent C. Lombardi, Ph.D.

Reviewer 1

The authors present a review article on a very interesting topic wherein the various constituents of mucosal immunity which may potentially impact the gut microbiota is discussed in detail. Authors have also given specific examples relating to few neuroimmune diseases wherein the modulations in immunological conditions are shown to confer a prominent effect on mucosal immunity. The topic chosen for the article is clinically relevant and it has been not particularly referred to in a lot of scientific articles. This also consequently shows a probable interest in the scientific reader community.

Thank you for your kind comments

General comments:
Many relevant scientific details have been discussed in detail however the manuscript shows an immense lack of organization and structure. Many of the introductory details presented by the authors have been already discussed in several research articles and the redundancy should have been avoided. Although the authors have attempted to demonstrate the relation between mucosal immunity and the gut-microbiota-brain axis, many of the subsections seem to drift away from the main focus of the article. This might cause the readers to get distracted in the course of reading and therefore it needs to be substantially changed. A review article should generally be very focused and should highlight novel findings relating to the main topic, rather than describing the already existing and well-known established facts in detail. A specific suggestion would be to concise the common facts relating to the subtopics and focus on more associated disease areas, which would also enhance the readability of the article.

With respect to the concern that the article lacks organization, our goal was to present the article in a hierarchical manner; for instance, starting with the background, then the architecture of mucosal immunity, followed by innate and then adaptive immunity, and closing with examples in the context of the neuroimmune diseases. A better way to organize the article was not obvious to us so to help improve the readability and addresses the reviewer's concern, we have made modifications to the entire article to make it more concise and easier to read (see amended manuscript). 

With respect to the reviewer's contention “A review article should generally be very focused and should highlight novel findings relating to the main topic, rather than describing the already existing and well-known established facts in detail” I would respectfully submit to the reviewer that there are many types of review papers that address a variety of topics. For instance, in the article by Palmatier et al. (2018)  Review articles: purpose, process, and structure1, it is stated that one purpose of a review article is to “Develop conceptual frameworks to reconcile and extend past research” which is what the present article endeavors to do. Although the reviewer’s definition of a review article is valid and probably the most common, there are clearly many different types of review articles that are written for a variety of reasons. As stated in our abstract “The goal of this review is to accommodate readers with limited backgrounds in immunology and/or anatomy and provide an overview of the components of mucosal immunity that impact the gut microbiota. We will then discuss how altered immunological conditions may shape the gut microbiota and consequently affect neuroimmune diseases.” Because the gut-microbiota-brain axis spans two disciplines of biology; neurology, and mucosal immunology, this subject represents a unique issue in that those who study neurology may not have the necessary immunology background to understand and design experiments that rely on mucosal immunology. Accordingly, our goal was to bridge this gap by providing a terse overview of mucosal immunity and then tie this up by providing an overview of how mucosal immunity impacts a select group of diseases that have established gut-microbiota-brain axis connections. In general, it was our belief that mucosal immunity has been a neglected topic with respect to the gut-microbiota-brain axis and we hoped to address this shortcoming. However, with that said, as the reviewer has suggested, we have attempted to remove any overly simple immunology concepts in order to make the article more concise and have instead directed the reader to the necessary resources that address this issue. We ask the reviewer for a little latitude to address this unique situation.

There needs to be scientific illustrations, high quality reproduced images from the published articles (with appropriate copyright permissions) and tables which can help in providing a concise version of certain sections. However, the current manuscript does not show adequate figures or tables. Therefore, the current version needs to be significantly revised.

With respect to the graphic images, we initially embedded 300 DPI figures into the manuscript, which may not have been clear so we have changed these to 600 DPI, (which is considered to be extra high resolution) for direct viewing and we have added the individual files as 600 DPI JPEG images as a zip attachment as requested by the publisher. With respect to the concern of copyright of the images, all images were generated using the computer program Biorender. This was indicated by the references that are given in the caption for each image. In addition, the appropriate copyright permissions for the use of Biorender and the adapted images were uploaded to the publisher submission portal at the time the manuscript was initially submitted. We have uploaded new permissions to reflect the higher resolution images.

Specific comments:
Few points of consideration in addition to the general comments mentioned above are given here:

  1. Abstract section, second last sentence – “the goal of this review is to…..gut microbiota”- this sentence might not be accurate for abstract and needs to be either eliminated/ rephrased.

The sentence has been amended to read: “The purpose of this review is to provide a brief overview of the components of mucosal immunity that impact the gut microbiota and then discuss how altered immunological conditions may shape the gut microbiota and consequently affect neuroimmune diseases, using a select group of neuroimmune diseases as examples.”

  1. Section 2- architecture of the gastrointestinal immune system – this section as indicated above shows a very high redundancy of details which have already been very well established and published in several articles. It is highly recommended to revise this entire section and provide the high-level details with concise subsections.

With respect to the general spirit of the article, it is necessary to include some background of mucosal immunity for those readers with limited knowledge of this subject; however, we have attempted to make the references sections more concise and put more emphasis on the critical aspects of mucosal immunity that impact the gut microbiota (please see amended manuscript).

  1. The resolution of figure 1 is very poor. Similarly figures 2 and 3 also lacks resolution. Copyright permissions need to be sought if these are reproduced images and should be indicated appropriately.

Please see the previous response

  1. Section 3.2, paragraph 2-last sentence (line number 264-265)-“PRRs can occur in many neuronal sites”- please provide adequate literature support by citing reference articles.

Four references have been added to address this.

  1. Section 3.4- one suggestion is to consider transforming the details given in this section such as subtypes of ILCs to a table. This helps to improve the ease in understanding and also makes the section succinct. Similarly, please consider usage of tables wherever appropriate.

Thank you for this suggestion. The information suggested to be included as a table is addressed by a figure in this section and upon a survey of the literature, this type of graphic with respect to innate lymphoid cells is the most common way of presenting these data: please see the following references as examples (Nature Reviews Immunology volume 17, pages 665–678 (2017), Human Cell volume 32, pages 231–239 (2019), Am J Transplant. 2015 Nov;15(11):2795-801. doi: 10.1111/ajt.13394) Because this illustration is meant to emphasize how a discrete group of cytokines cause innate immune cells to respond by producing another group of cytokines, which in turn change the effector function of T cells we felt that the best way to portray this was using the standard method. Also, given the reviewer's suggestion to include more figures, this would decrease the figure content; however, to accommodate the reviewer concern, we have added headers to each column of the figure and added detailed explanations regarding each column in the caption to make it more reader-friendly.

  1. Section 5 is relevant to the context of article. However, this section also seems to be very long with details which have been already established. The authors also have not shown any relevant figures relating to this section. Again, the usage of illustrations and tables need to be exploited as this is a review article.

As suggested by the reviewer, we have shortened this section to make it more concise and have added an additional illustration See attached manuscript). We have also added more concise descriptions to supplement the article.

  1. Section 6.1 on AD- lines 688-689- there is a single bracket that is open which hasn’t been closed. Please clarify.

This typo has been fixed at adding the closing bracket. (induced by administration of Aβ25-35 into the cerebral ventricles)

  1. What is the rationale behind choosing these four neuroimmune diseases specifically? How about similar evidence for other neuroimmune diseases?

The choice of the example neuroimmune disease was chosen based on the author’s respective expertise. We have added a sentence to the abstract to articulate this and added an explanation at the beginning of section 6 that further articulated the rationale behind the choice of diseases.

  1. Section 6- please consider giving figures of important results described in the subsections which can be reproduced figures from the cited papers (with permission).

Although an excellent idea, given the short time frame required by the journal to submit the revision, it would not be practical to obtain the necessary permission to include such images. Therefore, to accommodate this concern as much as possible, we have directed the reader to specific high quality reviews where appropriate. 

  1. Section 6.3- first paragraph on PD- as indicated before, these details are well established and need not be explained again. Authors can describe PD in 1-2 brief statements and then get into the details relating to the context of article.

      This section has been amended to reflect the reviewer’s suggestion

  1. Section 6.3- paragraph 2- many statements lack reference citations. Examples being lines 791, 801 etc. Similar case for line 819 in paragraph 4.

      The necessary references have been added to this section.

  1. Section 6.3, page 18, last paragraph- lines 862-872- please indicate appropriate reference articles.

      The necessary references have been added to this section.

  1. Section 7- conclusions- line 933- ‘manifest’ should be ‘manifested’.

      This has been fixed.

  1. There are several articles describing gut dysbiosis and related alterations causing the pathogenesis of Huntington’s disease (doi: 10.1093/braincomms/fcaa110, doi.org/10.3389/fimmu.2020.603594, doi.org/10.3389/fnins.2022.90220, doi.org/10.1016/j.isci.2021.103687 etc.). Authors should also consider adding this relevant section.

      We have added a short section to address Huntington’s disease and have utilized most of the suggested references as well as others.

  1. Similarly, multiple sclerosis is another disease area wherein there have been established implications of gut microbiota (https://doi.org/10.3389/fimmu.2022.785644). Please consider adding similar relevant and associated sections so that the major disease areas are covered.

      We have added a short section to address MS.

  1. There was not much evidence on microbial–host metabolism interactions in the context of neurodegenerative disorders. A relevant and recent 2022 paper by Gubert et al. https://doi.org/10.1093/gastro/goac017 can be included.

(1) (https://doi.org/10.1007),

Host metabolism was beyond the scope of this review, however, we have included the suggested review to the introduction of section 6. Human neuroimmune diseases with altered microbiomes

Reviewer 2 Report

The review is interesting and well structured. However, there are some corrections.

Minor comments:

1. The authors should better highlight the purpose of the review

2. Please refer to doi: doi.org/10.4110/in.2021.21.e20;  10.1007/s12035-018-1064-2; doi.org/10.3389/fimmu.2020.60417

3. the authors should fix all the figures in the review as poor in quality (the text inside is difficult to read) 

4.     The authors should better check the manuscript for any typographical errors.

Author Response

We have recited the reviewer’s comments below and have indicated the corrective actions below the corresponding comment below as underlined text. We have also used track changes in the original word document so to make it easier to follow.

We thank the reviewer for their articulated and insightful comments. We appreciate the time and effort it took to make these suggestions and believe your suggestions have made the manuscript much better and more readable.

Thank you,

Vincent C. Lombardi, Ph.D.

Reviewer 2

Minor comments:

  1. The authors should better highlight the purpose of the review

The abstract has been amended to better articulated the purpose of the review.

  1. Please refer to doi: org/10.4110/in.2021.21.e20;  10.1007/s12035-018-1064-2; doi.org/10.3389/fimmu.2020.60417

References to these articles have been incorporated into the manuscript

  1. the authors should fix all the figures in the review as poor in quality (the text inside is difficult to read)

With respect to the graphic images, we initially embedded 300 DPI figures into the manuscript, which may not have been clear so we have changed these to 600 DPI, (which is considered to be extra high resolution) for direct viewing and we have added the individual files as 600 DPI JPEG images as a zip attachment as requested by the publisher.

  1. The authors should better check the manuscript for any typographical errors.

The manuscript had been carefully proofed to address any typos that were missed upon the original submission.

Round 2

Reviewer 1 Report

Authors have done a good job in revising the manuscript based on the suggestions given in the previous review. The authors have attempted to answer all 16 questions. Many sections have been made concise and important points have been highlighted. Additional sections specifically on more diseases have been included. Additional figure and references have also been incorporated. Overall quality of the manuscript has improved considerably. I have a minor suggestion which can be easily addressed by the authors. 
- Section 6, paragraph 1 in page 15, lines 807-810: "However, several excellent..diseases." Authors could retain the first part of the sentence, however I would strongly recommend to omit the second part ('and we would refer the reader to these reviews..diseases') as the first part itself is self explanatory and it would sound scientifically more valid. 

- A comment on authors' response to question 9. The response does not seem to be satisfactory as reproducing few figures from already cited papers and acquiring copyright permissions would not take a longer time. Specifically, with all advancements now and most articles giving access to copyright permissions though platforms like creative common license, this is practically a click away. However, the current figures in the manuscript seem to be good at this point. But I just wanted to bring the above point to the authors' attention.

- For some reason, the figures still appear to be of low resolution.  I believe this is probably because of the pdf conversion process. Since authors have supplied high res figures separately, hopefully the production team would be able to take care of it. 

With the above being said, the article once incorporated with the minor suggestion above should be good enough for publication in IJMS. 

Author Response

Authors have done a good job in revising the manuscript based on the suggestions given in the previous review. The authors have attempted to answer all 16 questions. Many sections have been made concise and important points have been highlighted. Additional sections specifically on more diseases have been included. Additional figure and references have also been incorporated. Overall quality of the manuscript has improved considerably. I have a minor suggestion which can be easily addressed by the authors. 
- Section 6, paragraph 1 in page 15, lines 807-810: "However, several excellent..diseases." Authors could retain the first part of the sentence, however I would strongly recommend to omit the second part ('and we would refer the reader to these reviews..diseases') as the first part itself is self explanatory and it would sound scientifically more valid. 

Thank you for this suggestion. We have deleted the last part of the sentence so how it reads "However, several excellent reviews are available that provide a more comprehensive analysis of each respective disease."

- A comment on authors' response to question 9. The response does not seem to be satisfactory as reproducing few figures from already cited papers and acquiring copyright permissions would not take a longer time. Specifically, with all advancements now and most articles giving access to copyright permissions though platforms like creative common license, this is practically a click away. However, the current figures in the manuscript seem to be good at this point. But I just wanted to bring the above point to the authors' attention.

Thank you for this information; its good to know and we will consider this on future manuscripts

- For some reason, the figures still appear to be of low resolution.  I believe this is probably because of the pdf conversion process. Since authors have supplied high res figures separately, hopefully the production team would be able to take care of it. 

I believe you are correct.

With the above being said, the article once incorporated with the minor suggestion above should be good enough for publication in IJMS.